# Prediction of Nodal Metastasis in Lung Cancer Using Deep Learning of Endobronchial Ultrasound Images

**DOI:** 10.3390/cancers14143334

**Published:** 2022-07-08

**Authors:** Yuki Ito, Takahiro Nakajima, Terunaga Inage, Takeshi Otsuka, Yuki Sata, Kazuhisa Tanaka, Yuichi Sakairi, Hidemi Suzuki, Ichiro Yoshino

**Affiliations:** 1Department of General Thoracic Surgery, Graduate School of Medicine, Chiba University, Chiba 260-8670, Japan; yuki_ito_1989@yahoo.co.jp (Y.I.); potatolunch@yahoo.co.jp (T.I.); y.sata.0506@gmail.com (Y.S.); kazutanaka1118@yahoo.co.jp (K.T.); y_sakairi1@chiba-u.jp (Y.S.); hidemisuzukidesu@yahoo.co.jp (H.S.); iyoshino@faculty.chiba-u.jp (I.Y.); 2Department of General Thoracic Surgery, Dokkyo Medical University, Tochigi 321-0207, Japan; 3Advanced Image Processing Technology 4, Electrical Engineering, Olympus Medical Systems Corporation, Tokyo 192-8507, Japan; takeshi.otsuka@olympus.com

**Keywords:** EBUS-TBNA, echo B-mode imaging, deep learning-based computer-aided diagnosis, nodal staging

## Abstract

**Simple Summary:**

Endobronchial ultrasound-guided transbronchial aspiration is a minimally invasive and highly accurate modality for the diagnosis of lymph node metastasis and is useful for pre-treatment biomarker test sampling in patients with lung cancer. Endobronchial ultrasound image analysis is useful for predicting nodal metastasis; however, it can only be used as a supplemental method to tissue sampling. In recent years, deep learning-based computer-aided diagnosis using artificial intelligence technology has been introduced in research and clinical medicine. This study investigated the feasibility of computer-aided diagnosis for the prediction of nodal metastasis in lung cancer using endobronchial ultrasound images. The outcome of this study may help improve diagnostic efficiency and reduce invasiveness of the procedure.

**Abstract:**

Endobronchial ultrasound-guided transbronchial needle aspiration (EBUS-TBNA) is a valid modality for nodal lung cancer staging. The sonographic features of EBUS helps determine suspicious lymph nodes (LNs). To facilitate this use of this method, machine-learning-based computer-aided diagnosis (CAD) of medical imaging has been introduced in clinical practice. This study investigated the feasibility of CAD for the prediction of nodal metastasis in lung cancer using endobronchial ultrasound images. Image data of patients who underwent EBUS-TBNA were collected from a video clip. Xception was used as a convolutional neural network to predict the nodal metastasis of lung cancer. The prediction accuracy of nodal metastasis through deep learning (DL) was evaluated using both the five-fold cross-validation and hold-out methods. Eighty percent of the collected images were used in five-fold cross-validation, and all the images were used for the hold-out method. Ninety-one patients (166 LNs) were enrolled in this study. A total of 5255 and 6444 extracted images from the video clip were analyzed using the five-fold cross-validation and hold-out methods, respectively. The prediction of LN metastasis by CAD using EBUS images showed high diagnostic accuracy with high specificity. CAD during EBUS-TBNA may help improve the diagnostic efficiency and reduce invasiveness of the procedure.

## 1. Introduction

Endobronchial ultrasound-guided transbronchial aspiration (EBUS-TBNA) is a minimally invasive and highly accurate modality for the diagnosis of lymph node (LN) metastasis and is useful for pre-treatment biomarker test sampling in patients with lung cancer [1]. According to the current guidelines for lung cancer staging, EBUS-TBNA is recommended as the best first test for nodal staging prior to considering surgical procedures [2].

During EBUS-TBNA, multiple LNs are often encountered within the same nodal station. In this process, selecting the most suspicious LN for sampling is important, considering the difficulty of sampling all LNs using EBUS-TBNA under conscious sedation. Thus, EBUS image analysis is useful for predicting nodal metastasis; however, it can only be used as a supplemental method to tissue sampling [3]. We have previously reported the utility of six distinctive ultrasound and Doppler features on EBUS ultrasound images for predicting nodal metastasis [4,5]. However, categorization of image characteristics was not reliable owing to the fact it was subjective and varied significantly with the operator. Therefore, we sought an objective method to predict nodal metastasis. Elastography is a potential solution since it can visualize the relative stiffness of targeted tissues within the region of interest and helps to predict LN metastases. Moreover, it uses objective parameters such as a stiff area ratio [6,7]. However, elastography requires additional operations during the procedure, and its parameters do not reflect real-time values.

In recent years, deep learning (DL)-based computer-aided diagnosis (CAD) using artificial intelligence (AI) technology has been introduced in research and clinical medicine. CAD has been used for radiology, primarily in the areas of computed tomography (CT), positron emission tomography-CT (PET-CT), and ultrasound images, and for the diagnosis of several tumors, such as breast cancer and gastrointestinal tumors [8,9,10,11].

If real-time CAD-based prediction of nodal metastasis during EBUS-TBNA is made possible, the operator can easily identify the most suspicious node for diagnosis, thereby reducing the procedure time of EBUS-TBNA. The well-experienced EBUS operator could predict benign lymph nodes with approximately 90% accuracy by subjective categorization of EBUS ultrasound characters. The AI-CAD technology might make “the expert level prediction of nodal diagnosis” possible even for non-experts. The purpose of this study is to investigate the feasibility of CAD for the prediction of LN metastasis in lung cancer using endobronchial ultrasound images and DL technology. 

## 2. Materials and Methods

### 2.1. Participants

Patients with lung cancer or those suspected of suffering from lung cancer who underwent EBUS-TBNA for the diagnosis of LN metastasis were enrolled in this study. We prospectively collected clinical information and images related to bronchoscopy since April 2017 (registry ID: UMIN000026942), and the ethical committee allowed prospective case accumulation with written consent (ethical committee approval ID: No. 2563, Chiba University Graduate School of Medicine). The EBUS-TBNA video clips from April 2017 to December 2020 were retrospectively reviewed, and the patient’s clinical information was obtained from electronic medical records (ethical committee approval ID: No. 3538, Chiba University Graduate School of Medicine). This was a collaborative study between the Chiba University Graduate School of Medicine and Olympus Medical Systems Corp. (Tokyo, Japan). All patient identifiers were deleted, and the image data were sent to the Olympus Medical Systems Corp.’s laboratory and analyzed using DL technology (ethical committee approval ID: OLET-2019-008, Olympus Medical Systems Corp.). This study was conducted in accordance with the principles of the Declaration of Helsinki.

### 2.2. EBUS-TBNA Procedure

The patients underwent EBUS-TBNA under local anesthesia with moderate conscious sedation using midazolam and pethidine hydrochloride. OLYMPUS BF-UC290F and EU-ME1 and EU-ME2 PREMIER PLUS were used to observe LNs. Systematic nodal observation starting from the N1, N2, and N3 stations using B-mode, Doppler mode, and elastography was first performed. The size of each LN was measured, and EBUS-TBNA was performed for LNs > 3 mm along the short axis on the EBUS image. TBNA was initiated at N3, N2, and N1 stations to avoid overstating. For TBNA, a dedicated 22-gauge or 21-gauge needle (NA-201SX-4022, NA-201SX-4021, Olympus Medical Systems Corp., Tokyo, Japan) was used, and rapid on-site evaluation was performed during the procedure. All EBUS procedures were performed by skilled operators (T.N. and Y.Sakairi.) or under their supervision.

### 2.3. Confirmation Diagnosis of EBUS-TBNA

Rapid on-site evaluation by DiffQick staining and conventional cytology by Papanicolaou staining were performed and diagnosed by a cytopathologist. The histological core was collected in CytoLyt solution and fixed in 10% neutral buffered formalin. The formalin-fixed paraffin-embedded specimens were stained with hematoxylin and eosin (H&E) and subjected to immunohistochemistry. Cytology as well as histology was evaluated by independent pathologists who provided pathological diagnosis [12]. The referenced final diagnoses were as follows: (1) malignant cells were proven by EBUS-TBNA, (2) histological diagnosis was made for surgically resected samples after EBUS-TBNA, (3) clinical follow up by radiology after 6 months. 

### 2.4. EBUS Image Extraction and Image Data Sets

Ultrasound images were recorded as video clips in the MP4 format; divided into shorter clips featuring each LN using video editing software, XMedia Recode 3.4.3.0 (Sebastian Dörfler, Eschenbergen, Germany); and subsequently anonymized using the dedicated software VideoRectFill (Olympus Medical Systems Corp.). All patient information was manually masked on the software. An anonymized video clip was provided to Olympus Medical Systems Corp. with diagnostic information linked to each LN.

In this study, we retrospectively and prospectively collected cases and investigated the detection of LN metastasis in each LN. The evaluation methods are illustrated in Figure 1. We retrospectively and prospectively collected LNs. We attempted both five-fold cross-validation and hold-out methods for evaluation. Because the images from the video clips included different ultrasound processors (EU-ME1 and EU-ME2 PREMIER PLUS) and different image sizes, these images were allocated equally to each training, validation, and testing group (Appendix A).

### 2.5. Adjustment of Images for DL

Prior to image analysis, the videos were decomposed into time-series images, from which images of different scenes were extracted. The areas in which the B-mode was drawn were cropped from the images and the cropped images were resized to the same size. To increase the generalizability of the DL algorithm, data augmentation was applied only to the training images, and the number of training images was increased. Scaling and horizontal flipping were used in the data augmentation process.

### 2.6. DL Algorithm Design

The Convolutional Neural Network (CNN) structure used in this study for LN metastasis detection is shown in Appendix A. The metastasis detection CNN comprises a feature extraction CNN and detection CNN. The feature extraction CNN comprises multiple stages with each stage having multiple blocks and one downsampling layer. The final stage did not include a downsampling layer. We used the Xception block for each block [13]. The downsampling layer comprises two or more strides of the convolution layer. The detection CNN comprises two convolution layers: one for classification and another for positioning.

Initially, the ultrasound image was input to the feature CNN, and local features, such as edges and textures, were extracted from the input image in the first block. As it progressed through the network, its features were integrated and finally converted into features useful for detection.

Subsequently, the features useful for metastasis detection were input into the detection CNN. The detection CNN outputs the probability and bounding box coordinates and sizes for both metastasis and nonmetastasis. The bounding box with the highest probability was selected from among all the metastatic and non-metastatic bounding boxes in the sequence. Finally, the metastasis or non-metastasis parameters, coordinates and size of the bounding box were obtained as the detection result.

### 2.7. Five-Fold Cross-Validation Method and the Hold-Out Method

For the five-fold cross-validation method, 80% of all the images were used for training and validation. The images were divided into five sections: four sections were used for training, and the last section was used for validation. By changing the validation section, the training and validation were repeated five times. The prediction yield was calculated as the average of the results of each validation.

In the hold-out method, all images were used for training and testing. All of the images comprising the 80% used for the five-fold cross-validation method were used for training. The remaining 20% of the images that were not used for the five-fold cross-validation method were used for testing, following which the prediction yield was calculated. 

The images of different sizes from the two ultrasound scanners (EU-ME1 and EU-ME2 PREMIER PLUS) were allocated proportionately in each section to avoid selection bias.

### 2.8. Statistical Analysis

The “Image” represents “per image” basis analysis and the “Lymph node” represents “per lymph node” basis analysis. The “per image” analysis was based on the accuracy of nodal metastasis prediction for each image. Due to limited number of still images, we used the video clips for analysis. However, in this case, multiple images with varying ultrasound features were included for each targeted lymph node, resulting in variation in the judgement of the AI-CAD system. Therefore, in addition to “per image” analysis, we included “per lymph node” analysis in which multiple images were evaluated for each lymph node. The “per lymph node” analysis included (1) calculation of the ratio between the number of images judged benign and malignant, (2) predicting as benign or malignant based on the ratio >50%, (3) analysis of the accuracy of nodal metastasis prediction for each lymph node. 

Sensitivity, specificity, positive predictive value, negative predictive value, and diagnostic accuracy were calculated using standard definitions. Statistical analysis was performed using Fisher’s exact test and chi-square test for categorical outcomes, and Student’s t-test for continuous variables. Data were analyzed using the JMP Pro 15 software (SAS Institute Inc., Cary, NC, USA). Statistical significance was set at *p* < 0.05. 

## 3. Results

Ninety-five cases with a total of 170 LNs were enrolled in the study. Two cases (two LNs) were excluded because of a history of malignant lymphoma. Cases of large-cell carcinoma and large-cell neuroendocrine carcinoma (one LN each) were also excluded because they could not be assigned to both the training and testing sets. Finally, 91 cases and 166 LNs were analyzed in this study (Figure 2). In this cohort, 64 LNs (38.5%) were diagnosed as metastatic and 102 LNs (61.5%) as non-metastatic by pathology. The characteristics of the enrolled patients and LNs are listed in Table 1.

Pathological diagnosis including cytology and histology were performed for all lymph nodes. The success rate of each diagnosis was shown in Table 2. For adenocarcinoma cases, molecular biomarker testing was performed for selected cases. For non-small cell lung cancer cases, evaluation for PD-L1 (22C3) immunohistochemistry was done for selected cases. Each success rate, detection rate, and testing rate was shown in Table 2.

First, we evaluated the ability of AI-CAD to detect LN metastasis using endobronchial ultrasound images. A total of 5255 and 6444 extracted images from the video clip were analyzed using the five-fold cross-validation and the hold-out methods, respectively (Figure 1). The representative EBUS images judged by AI-CAD in this study are shown in Appendix A.

Using the five-fold cross-validation method, the LN-based diagnostic accuracy, sensitivity, specificity, positive predictive value, and negative predictive value of the AI-CAD were measured to be 69.9% (95% CI, 32.4–75.2%), 37.3% (95% CI, 27.8–49.1%), 90.2% (95% CI, 82.9–92.3%), 70.4%, and 69.8%, respectively (Figure 3). However, although the specificity was high, the sensitivity of this method was low.

Using the hold-out method, the LN-based diagnostic accuracy, sensitivity, specificity, positive predictive value, and negative predictive value of the AI-CAD were measured to be 87.9% (95% CI, 75.4–94.1%), 76.9% (95% CI, 58.9–92.9%), 95.0% (95% CI, 79.3–100%), and 90.9% and 86.4%, respectively (Figure 4). 

Regarding the diagnostic yield by lung cancer subtypes, the diagnostic accuracy rates were 90.5% for no malignancy, 76.9% for adenocarcinoma, 61.1% for squamous cell carcinoma, and 93.9% for small cell lung cancer (Figure 5).

## 4. Discussion

The potential applications of AI technology are rapidly growing in the medical field and are expected to facilitate the demanding work of medical staff. The concept of AI, including such systems as machine learning and DL, has been growing in popularity since the evolution of graphics processing units. AI-CAD is one of the AI applications that has been actively developed in radiology. Significant work has been done in the area of combining radiomics and AI-CAD technology, which helps support the diagnosis of benign and malignant tumors, prediction of histology, stage, genetic mutations, and prediction of treatment response and recurrence using CT and PET-CT images [14,15,16,17,18]. AI-CAD is highly useful in analyzing huge amounts of extracted information that includes information invisible to humans. AI-CAD produces objective indicators based on the judgment, knowledge, and experience of experts. During EBUS-TBNA, a highly skilled operator can select the most suspicious LN to sample, based on a subjective categorization of ultrasound image characteristics. In contrast, by applying AI-CAD technology in EBUS, even a trainee can easily select the target LN for sampling, in addition to the dual advantages of a more efficient and less invasive procedure. In this study, we used the CNN algorithm with Xception to predict nodal metastasis based on the ultrasound images of LNs. Using the hold-out method, AI-CAD exhibited a feasible diagnostic accuracy of 84.7%, on average, per LN basis. In this study, the combination of Xception and the hold-out method resulted in the highest diagnostic yield.

The comparison between the five-fold cross-validation and the hold-out methods, demonstrated that the hold-out method exhibited a superior diagnostic yield in this study setting. First, we evaluated using five-fold cross-validation, and then used hold-out method as the standard for developing AI-CAD technology. The number of evaluated images was increased by 20% for the hold-out method compared to five-fold cross-validation. The increased number of images helped with comprehensive covering of image variation and contributed toward better AI-CAD accuracy. The images used in this study were obtained using two different ultrasound image processors (EU-ME1 and EU-ME2 PREMIER PLUS). In addition, a certain amount of collected images (approximately 10% of all images) were of different sizes owing to the different screen sizes of the various video clips. These variations might affect the diagnostic yield of the five-fold cross-validation and the hold-out methods. Thus, for the analysis of different-size images the image had to be resized and then analyzed, which resulted in an adversarial example (AE). An AE is an event in which AI misrecognizes an image as completely different data owing to the addition of insignificant noises that are imperceptible to humans. [19] Therefore, in this study, these problems were solved by allocating images of different sizes in equal proportions for AI-CAD analysis. 

The final diagnostic accuracy and specificity for the prediction of LN metastasis using AI-CAD in this study were 87.9% and 95.0%, respectively. Previous studies have reported comparable but lower values. For instance, Ozcelik et al. reported an accuracy rate of 82% and specificity of 72% for the diagnosis of lung cancer LN metastasis in 345 LNs by CNN using MATLAB [20]. Churchill et al. reported an accuracy rate of 72.8% and a specificity of 90.7% for the diagnosis of lung cancer LN metastasis in 406 LNs by CNN using NeuralSeg [21]. It is noteworthy, however, that the specificity of CNN-based diagnosis for the prediction of nodal metastasis was found to be high, and this might help avoid futile biopsies and reduce examination time as well as the risk of co-morbidities. 

Furthermore, we examined the diagnostic yield of lung cancer subtypes (Figure 5). The diagnostic yield was highest for small cell lung cancer, while the accuracy rate was relatively low for squamous cell carcinoma. Squamous cell carcinoma is often accompanied by signs of coagulation necrosis at the center of the LN, which might affect diagnostic accuracy.

In this study, the prediction rate for squamous cell carcinoma was relatively lower than other histology. One of the possible reasons of this phenomenon was that the squamous cell carcinoma often shows various histological characters, such as necrosis and fibrosis, and it reflects the characters on an EBUS ultrasound image, such as necrosis sign and heterogeneity of echogram. These various ultrasound image features might cause difficulties for learning and validation by AI-CAD, resulting in a lower prediction rate. Although better AI-CAD analysis required more numbers of squamous carcinoma cases for comprehensive coverage of the image variation of squamous cell carcinoma, the number of actual squamous cell carcinoma cases were relatively low in this study. If we could increase the number of squamous cell carcinoma cases, the diagnostic yield could be better in the future.

This study has several limitations. First, the study population was limited, and we used video clips to overcome the limitations of the small sample size. Some cases underwent multiple LN assessments, and multiple LN images were obtained from a single case, which might show similar image characteristics. Second, we used only B-mode images in this study. Several reports have demonstrated the utility of other imaging modalities such as Doppler mode imaging and elastography [5,6]. Finally, Xception was used for the CNN in this study, although there is currently no consensus as to which algorithm should be used to analyze echo images. To develop the optimal method of AI-CAD for EBUS imaging, a larger prospective cohort study is required in the future. In addition, AI-CAD diagnosis using other imaging modalities such as Doppler mode and elastography should be examined to improve the diagnostic yield of AI-CAD for EBUS imaging. 

In this study cohort, the prevalence of nodal metastasis was 38.5%, which was relatively low in comparison with the previous report. Most of the enrolled patients were referred to the surgical department as resectable lung cancer patients. In real clinical setting, the AI-CAD technology will be useful if the operator cannot decide which one to be sampled during EBUS-TBNA. The operator would not need the image analysis support for selecting the target when the lymph node is obviously enlarged. Thus, this study demonstrated that the AI-CAD can be used to support the nodal staging for surgically treatable patients.

## 5. Conclusions

In this study, we found that AI-CAD (a combination of Xception and the hold-out method) for the prediction of LN metastasis using endobronchial ultrasound images is feasible and exhibits high diagnostic accuracy and specificity. AI-CAD for EBUS may reduce futile biopsies of LNs, shorten examination time, and make EBUS-TBNA a less invasive procedure, regardless of operator experience. 

## Figures and Tables

**Figure 1 cancers-14-03334-f001:**
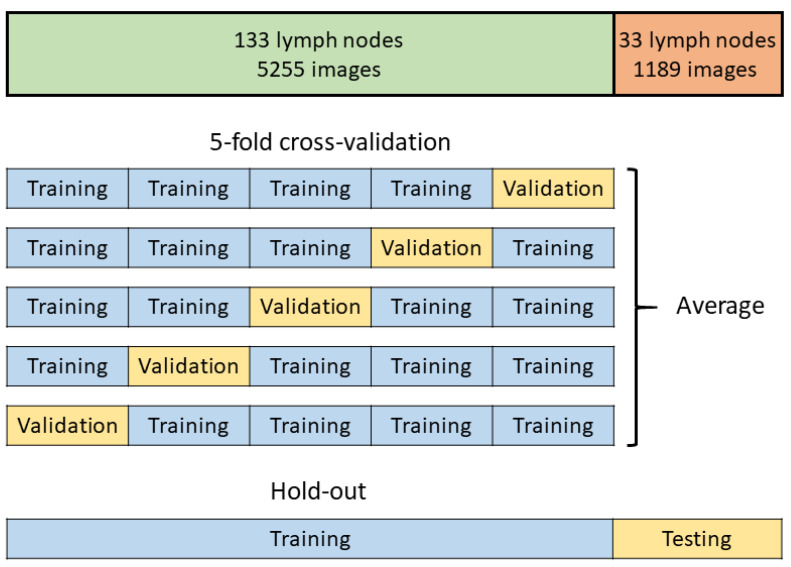
The concept of deep learning algorithm.

**Figure 2 cancers-14-03334-f002:**
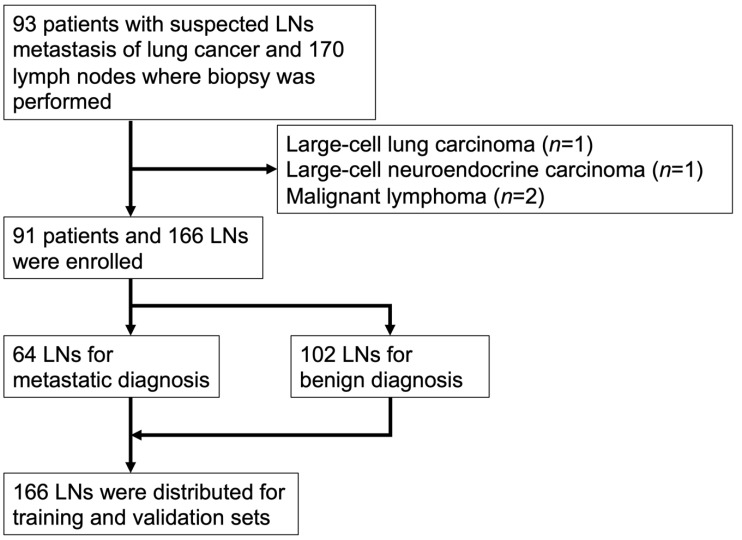
Study cohort flow chart. One hundred sixty-six lymph nodes and 6444 images from 91 patients were enrolled in the final analysis.

**Figure 3 cancers-14-03334-f003:**
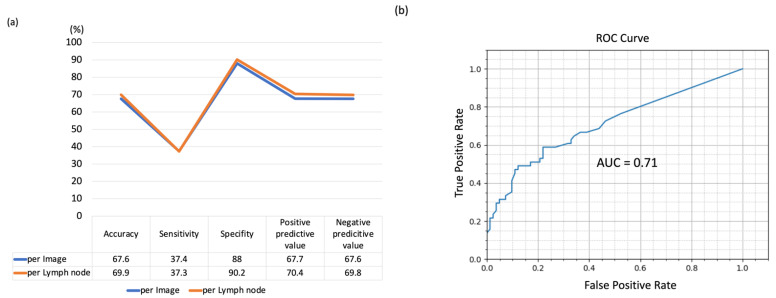
The result of AI-CAD lung cancer lymph node diagnosis accuracy analysis using echo images by five-fold cross validation method. (**a**) Diagnostic yield by per image basis and per lymph node basis. (**b**) ROC curve.

**Figure 4 cancers-14-03334-f004:**
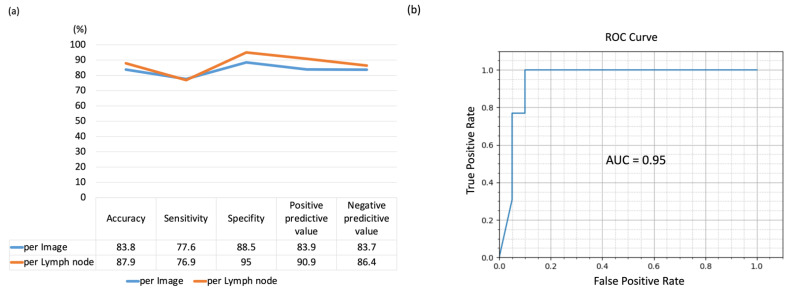
The result of AI-CAD lung cancer lymph node diagnosis accuracy analysis using echo images by hold-out method. (**a**) Diagnostic yield by per image basis and per lymph node basis. (**b**) ROC Curve.

**Figure 5 cancers-14-03334-f005:**
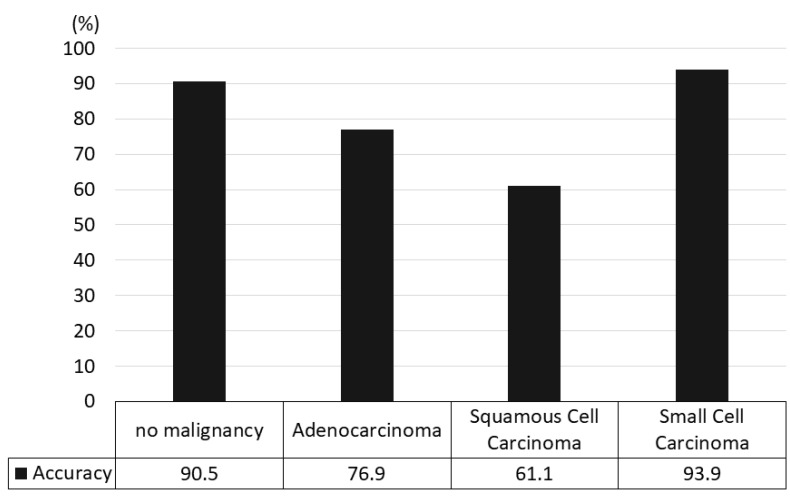
The accuracy rates of the hold-out method by lung cancer subtype.

**Table 1 cancers-14-03334-t001:** Patients’ and nodal characteristics.

No. of patients	91
Age (y) (median, range)	74 (12–86)
Gender	
male	61 (67.0%)
female	30 (33.0%)
No. of lymph nodes	166
Diagnosis	
Metastatic lymph nodes	64 (38.5%)
Adenocarcinoma	40 (24.0%)
Squamous cell carcinoma	15 (9.0%)
Small cell carcinoma	9 (5.4%)
Benign lymph nodes	102 (61.5%)
Lymph node station	
1	1
2R	13
3p	2
4R/4L	41/25
7	43
8	1
10R/10L	5
11s/11i/11(Lt.)	15/6/4
12	9
13	1
Lymph node size of long axis	Average (range), mm
All lymph nodes	12.9 (3.0–29.2)
Metastatic lymph nodes	15.5 (3.0–29.2)
Benign lymph nodes	11.3 (3.5–21.8)

**Table 2 cancers-14-03334-t002:** Detailed results of pathological diagnosis and biomarker testing in this study.

Metastatic Lymph Node (*n* = 64)	Diagnosed by Cytology	Diagnosed by Histology	Success Rate of Molecular Testing	Detection Rate of Driver Gene Mutations	Testing for PD-L1 Immunohistochemistry
Adenocarcinoma (*n* = 40)	37/40 (92.5%)	37/40 (92.5%)	22/24 (91.7%)	13/22 (59.0%)	22/40 (55.0%)
Squamous cell carcinoma (*n* = 15)	13/15 (86.7%)	14/15 (93.3%)	N/A	N/A	7/15 (46.7%)
Small cell carcinoma (*n* = 9)	9/9 (100%)	9/9 (100%)	N/A	N/A	N/A

## Data Availability

Not applicable.

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
