# Peer review of "Prediction of Nodal Metastasis in Lung Cancer Using Deep Learning of Endobronchial Ultrasound Images"

_cancers, 2022, doi:10.3390/cancers14143334_

Round 1

Reviewer 1 Report

The paper reports a study evaluating the use of deep learning of endobronchial ultrasound images for predicting nodal metastasis in lung cancer.

The present study is well-written and well-presented providing some novel information in the field of diagnosis NSCLC metastasis. However, the authors didn't discuss about the importance to obtain tissue for the right diagnosis of lung cancer subtype of as well as for molcular analysis that is mandatory for target therapy of NSCLC. This is a limit of the study that the authors should be discuss and they will add a paragraph in the results section regarding the histology and feasibility of molecular analysis on the material obtained with EBUS-TBNA. 

Reviewer 2 Report

The authors investigated the prediction accuracy of LN metastasis in lung cancer using EBUS images and deep learning (DL) technology in 91 cases (166 LNs). This DL was evaluated using both the 5-fold cross-validation and hold-out methods. Finally, the authors showed that the diagnostic accuracy and specificity for the prediction of LN metastasis using AI-CAD by hold-out method were 87.9% and 95.0%, respectively.

Generally, EBUS procedure is an essential step in the LN staging of lung cancer. In recent years, DL is one of the fastest-growing areas in medical image analysis and has had a significant impact on different applications, both clinical and research. From this point of view, the present study is considered a very interesting and well-written manuscript. Therefore, this manuscript needs some supplement points to improve the quality of the article and the understanding of international readers. 

1. In this cohort, 64 LNs (38.5%) were diagnosed as metastatic and 102 LNs (61.5%) as non-metastatic by pathology. The malignancy rate is relatively low. These ratios may also affect the results. Further discussion of these limitations is needed.

2. What is the difference between image and lymph node results in figure 3 and 4?

3. It would be helpful for understanding of international readers and clinicians to show representative EBUS images. 

4. In methods, “~ EBUS-TBNA was performed for LNs > 5 mm along the short axis on the EBUS image”. However, the range of LN size in table 1 is less than 5mm. Why?

Also, what do NA8, NA1, and NA7 in table 1 mean?

5. There is a difference in the diagnostic yield by lung cancer subtypes (figure 5). Please comment in more detail on the reasons for these differences, possible theoretical backgrounds, or the opinions of the authors.

Reviewer 3 Report

General comments:

This study investigated diagnostic accuracy and specificity of AI-CAD related to B-mode images during EBUS. This theme is one of the topics recently focused. However, I think this study have some problems, therefore, several modifications are necessary.

Comment 1:

The authors should show the primary outcome, which was decided when this study was planned, in the method section. Did you have any specific threshold goals that you wanted to prove in this study, such as a diagnostic accuracy that would be useful in daily clinical practice? If you did not any specific goals, I think that the comparison to the conventional subjective expert operator’s EBUS image analysis should be considered.

Comment 2:

The authors should add a consideration as to why the results between the 5-fold cross-validation and the hold-out methods is different. The difference of these results was predicted? The rationale for comparing the two methods should be clearly stated.

Comment 3:

In general, I think the authors should show the ROC curves and AUC in Figure 3 and Figure 4 to demonstrate the usefulness of this AI model. The line graphs in Figure 3 and Figure 4 were useless.

Comment 4:

In Figure 3 and Figure 4, I think it is difficult to understand the difference between two groups indicated “Image” and “Lymph node”. Please provide a clear explanation for figure legend.

Comment 5:

The authors discussed why the accuracy rate was relatively low for squamous cell carcinoma in discussion section as following, “Squamous cell carcinoma is often accompanied by signs of coagulation necrosis at the center of the LN, which might affect diagnostic accuracy.” However, if the signs of coagulation necrosis are present, the lymph nodes would be easier to distinguish from benign lymph nodes. Therefore, additional discussion would be needed.

Comment 6:

In Table 1, is one of the “11s” notations a mistake for “11i”?

In Table 1, is lymph node size short axis or long axis?

In Table 1, what means “NA 8”, “NA 1”, and “NA 7”?

Author Response

Response to Reviewer 3 Comments

General comments:

This study investigated diagnostic accuracy and specificity of AI-CAD related to B-mode images during EBUS. This theme is one of the topics recently focused. However, I think this study have some problems, therefore, several modifications are necessary.

Comment 1:

The authors should show the primary outcome, which was decided when this study was planned, in the method section. Did you have any specific threshold goals that you wanted to prove in this study, such as a diagnostic accuracy that would be useful in daily clinical practice? If you did not any specific goals, I think that the comparison to the conventional subjective expert operator’s EBUS image analysis should be considered.

Response 1: Thank you for your valuable suggestion. In this study, the primary outcome was diagnostic yield by AI-CAD; however, there were no specific threshold goals because there was no standard diagnostic yield for EBUS image analysis by AI-CAD technology and this was a feasibility study. As the reviewer mentioned, the well experienced experts could predict benign lymph nodes with approximately 90% accuracy by subjective categorization from previous reports. We have added the following sentence in Introduction to address your concerns.

 “The well experienced EBUS operator could predict benign lymph nodes with approximately 90% accuracy by subjective categorization of EBUS ultrasound characters. The AI-CAD technology might make “the expert level prediction of nodal diagnosis” possible even for non-experts.”

Comment 2:

The authors should add a consideration as to why the results between the 5-fold cross-validation and the hold-out methods is different. The difference of these results was predicted? The rationale for comparing the two methods should be clearly stated.

Response 2: Thank you for bringing this to our attention. Accordingly, we have added the following explanation in Discussion.

“First, we evaluated using five-fold cross-validation, and then used hold-out method as the standard for developing AI-CAD technology. The number of evaluated images was increased by 20% for hold-out method than five-fold cross-validation. The increased number of images helped in comprehensive covering of image variation, and contributed toward better AI-CAD accuracy.”

Comment 3:

In general, I think the authors should show the ROC curves and AUC in Figure 3 and Figure 4 to demonstrate the usefulness of this AI model. The line graphs in Figure 3 and Figure 4 were useless.

Response 3: Thank you for the valuable suggestions. We have added the ROC curve in Figures 3 and 4.

Figure 3 The result of AI-CAD lung cancer lymph node diagnosis accuracy analysis using echo images by 5-fold cross validation method. (a) Diagnostic yield by per image basis and per lymph node basis. (b) ROC curve.

Figure 4 The result of AI-CAD lung cancer lymph node diagnosis accuracy analysis using echo images by Hold-out method. (a) Diagnostic yield by per image basis and per lymph node basis. (b) ROC Curve.

Comment 4:

In Figure 3 and Figure 4, I think it is difficult to understand the difference between two groups indicated “Image” and “Lymph node”. Please provide a clear explanation for figure legend.

Response 4: In accordance with your comment, the labels were changed as “per image” and “per lymph node” and the following explanation was added to clarify the statistical analysis.

 “The “Image” represents “per image” basis analysis and the “Lymph node” represents “per lymph node” basis analysis. The “per image” analysis was based on the accuracy of nodal metastasis prediction for each image. Due to limited number of still images, we used the video clips for analysis. However, in this case, multiple images with varying ultrasound features were included for each targeted lymph node, resulting in variation in the judgement of the AI-CAD system. Therefore, in addition to “per image” analysis, we included “per lymph node” analysis in which multiple images were evaluated for each lymph node. The “per lymph node” analysis included 1) calculation of the ratio between the number of images judged benign and malignant, 2) predicting as benign or malignant based on the ratio >50%, 3) analysis of the accuracy of nodal metastasis prediction for each lymph node.”

Comment 5:

The authors discussed why the accuracy rate was relatively low for squamous cell carcinoma in discussion section as following, “Squamous cell carcinoma is often accompanied by signs of coagulation necrosis at the center of the LN, which might affect diagnostic accuracy.” However, if the signs of coagulation necrosis are present, the lymph nodes would be easier to distinguish from benign lymph nodes. Therefore, additional discussion would be needed.

Response 5: Thank you for the insightful comments.

Following your suggestions, we have added the following explanation in Discussion of the revised manuscript.

“In this study, the prediction rate for squamous cell carcinoma was relatively lower than other histology. One of the possible reasons of this phenomenon was that the squamous cell carcinoma often shows various histological characters such as necrosis and fibrosis and it reflect the characters on EBUS ultrasound image such as necrosis sign and heterogeneity of echogram. These various ultrasound image features might made difficult for learning and validation by AI-CAD which resulting in lower predicting rate. Although better AI-CAD analysis required more numbers of squamous carcinoma cases for comprehensive coverage of image variation of squamous cell carcinoma, the number of actual squamous cell carcinoma cases were relatively low in this study. If we could increase the number of squamous cell carcinoma cases, the diagnostic yield could be better in the future.”

Comment 6:

In Table 1, is one of the “11s” notations a mistake for “11i”?

Response 6: We have corrected it as “11i”.

In Table 1, is lymph node size short axis or long axis?

Response 7: The size in Table 1 was the lymph node size of short axis. It has been clarified now.

In Table 1, what means “NA 8”, “NA 1”, and “NA 7”?

Response 8: We have revised Table 1 to ensure clarity.
